# Circular Animal Protein Hydrolysates: A Comparative Approach of Functional Properties

**DOI:** 10.3390/antiox14070782

**Published:** 2025-06-25

**Authors:** Marta Monteiro, Luciano Rodrigues-dos-Santos, Andreia Filipa-Silva, Diana A. Marques, Manuela Pintado, André Almeida, Luisa M. P. Valente

**Affiliations:** 1CIIMAR/CIMAR LA, Centro Interdisciplinar de Investigação Marinha e Ambiental, Universidade do Porto, Av. General Norton de Matos S/N, 4450-208 Matosinhos, Portugal; lers.eduardo.97@gmail.com (L.R.-d.-S.); andreia.silva@ciimar.up.pt (A.F.-S.); dmarques@ciimar.up.pt (D.A.M.); 2Instituto de Ciências Biomédicas Abel Salazar, Universidade do Porto, Rua de Jorge Viterbo Ferreira, 228, 4050-313 Porto, Portugal; 3Laboratório Associado, Escola Superior de Biotecnologia, Centro de Biotecnologia e Química Fina (CBQF), Universidade Católica Portuguesa, Rua Diogo Botelho 1327, 4169-005 Porto, Portugal; 4ETSA, Empresa Transformadora de Subprodutos, 2660-119 Loures, Portugal; 5SEBOL, Comércio e Indústria do Sebo, S.A. (ETSA Group), 2660-119 Loures, Portugal

**Keywords:** circular bioeconomy, protein hydrolysis, bioactive peptides, animal nutrition, functional ingredients

## Abstract

The growing demand for nutraceuticals has driven interest in upcycling low-value proteins from processed animal by-products and insect larvae into functional protein hydrolysates. This study evaluated five such hydrolysates in comparison to a high-value commercial reference (CPSP90), assessing the proximate composition, amino acid profile, molecular weight distribution, antioxidant activity, and bacterial growth dynamics. Results revealed a wide variability in the composition and bioactivity, driven by the raw material and processing conditions. All hydrolysates displayed a medium to high crude protein content (55.1–89.5% DM), with SHARK being the most protein-rich. SHARK and SWINE hydrolysates were particularly rich in collagenic amino acids, while FISH and CPSP90 contained higher levels of essential amino acids. FISH and INSECT demonstrated the strongest antioxidant activity, with INSECT also showing the highest protein solubility. INSECT and SWINE further displayed mild, selective antibacterial effects, indicating a potential for disease mitigation. Conversely, SHARK and FISH supported opportunistic bacteria growth, suggesting a potential use as nitrogen sources in microbial media. These findings highlight the nutritional and functional versatility of animal-derived protein hydrolysates and support their integration into sustainable feed strategies within a circular bioeconomy.

## 1. Introduction

Protein hydrolysates have gained significant attention in recent years for their ability to deliver high-quality peptides with nutritional and health-promoting benefits in both human and animal nutrition [1]. Among their bioactive properties, their antioxidant potential is particularly noteworthy [2], along with their capacity to modulate microbial growth and anti-inflammatory responses, making them versatile functional ingredients [3,4]. Among the commercially available hydrolysates, CPSP90 (soluble fish protein concentrate) is widely recognized as a benchmark in animal nutrition, particularly in aquaculture, due to its high digestibility and palatability [5,6,7,8]. It is often used as a positive control or reference ingredient in experimental diets, especially when evaluating novel protein sources for fish feed [9,10,11] or simply to ensure feed acceptance and palatability [12,13].

Beyond their nutritional value and bioactive properties, protein hydrolysates may offer a sustainable solution for waste valorization, particularly within the framework of a circular economy. Regardless of how quickly or significantly an animal grows, only 30–40% of its body is typically utilized for human consumption [14,15]. The remainder generates substantial volumes of solid and liquid waste, classified in Europe as category 3 animal by-products that may be used for animal feed (Regulation (EC) No 1069/2009 and Regulation (EU) No 142/2011). With the anticipated global increase in the consumption of animal protein products, the associated waste will also rise, underscoring the urgent need for resource efficiency. In a world with finite resources, minimizing, recovering, and utilizing these by-products is critical. Failing to do so results in not only a considerable loss of potential revenue but also increased disposal costs [1]. To address this, the food and feed industries have adopted various technologies to repurpose residual raw materials into value-added products. While some by-products are already used in feed formulations, such as blood meal, poultry by-product meal, or feather meal [14] through hydrolysis, these protein-rich by-products can be transformed into novel functional ingredients, upcycling otherwise low-value raw materials. Hydrolysis provides an innovative approach to convert these protein-rich materials into functional ingredients [16,17]. This process effectively upcycles low-value raw materials, unlocking their potential for broader applications. Building on this, extensive research has focused on developing novel hydrolysis methods for meat and fish by-products to enhance the recovery of bioactive peptides and maximize their functional properties [16,18,19]. This is particularly relevant as animal production increases to meet the protein demand [20], presenting challenges such as increased stress in production systems, climate change, and a corresponding surge in disease outbreaks [21,22]. Incorporating functional ingredients like protein hydrolysates into animal diets offers a practical solution to these issues by helping to mitigate oxidative stress, strengthen immune resilience, and improve overall animal health under increasingly unpredictable environmental conditions [23,24,25].

A deeper understanding of protein hydrolysates and their functional properties across diverse raw materials is crucial for optimizing their applications in animal nutrition. This study aimed to evaluate the functional and bioactive properties of five protein hydrolysates derived from insects and by-products from sharks, fish, and porcine, using CPSP90 as a reference. The hydrolysates were analyzed for their proximate composition, amino acid profiles, molecular weight distribution, and solubility, with a particular emphasis on their antioxidant potential and ability to influence the proliferation of pathogenic bacteria relevant to aquaculture. These findings highlight the potential of locally produced animal by-products to develop innovative and sustainable functional ingredients that not only match but outperform a commercially established hydrolysate like CPSP90, while simultaneously addressing agro-food waste reduction.

## 2. Materials and Methods

### 2.1. By-Products Source and Hydrolysate Production

Locally sourced category 3 animal by-products, from shark, fish, swine, and insect larvae meal, were processed into five innovative protein hydrolysates. The processing of the shark, fish, insect, and swine by-products was performed by ETSA-SGPS, S.A. under confidential conditions, following EU regulations (Regulation (EC) No 1069/2009 and No 142/2011). The shark hydrolysate (SHARK) was derived from blue shark skin (*Prionace glauca*), a by-product from the filet processing industry (Brasmar—Trade Food S.A., Guidões, Portugal). This raw material underwent grinding, heat pre-treatment with hot water, and enzymatic hydrolysis, for 2 h, using Alcalase 2.4 L Pure (Novozymes, Bagsværd, Denmark) at 65 °C; pH 7–9. After enzyme inactivation at 90 °C, for 10 min, the mixture was sifted (450 µm) to remove small particles and non-hydrolyzed materials. The remaining liquid fraction was centrifuged at 5394× *g* to separate the protein, lipid, and mineral phases. The mineral phase was isolated immediately, while the protein and lipid phases were separated using a decanting funnel. The protein fraction was concentrated via rotary evaporation and subsequently spray-dried using a Büchi B-290 Spray Dryer (Büchi, Flawil, Switzerland). The fish hydrolysate (FISH) was produced from fish by-products (multispecies) obtained from various retailers and distributors in the fish processing and canning industries. These materials were minced and received heat pre-treatment similarly to SHARK. Then, the resulting mixture was hydrolyzed using Alcalase 2.4 L Pure, following a similar protocol as SHARK but for 4 h. After enzyme inactivation, the mixture was sifted and centrifuged to separate the phases. The protein fraction was concentrated and spray-dried, which is consistent with the process used for SHARK. The insect hydrolysate (INSECT) was prepared using black soldier fly larva meal (*Hermetia illucens*) obtained from Hermetia Baruth GmbH, Baruth/Mark, Germany. During initial processing, a portion of the fat content was removed from the larvae to create a defatted meal suitable for hydrolysate production. The raw material was pre-treated with hot water and hydrolyzed as described for SHARK but for 4h. Enzyme inactivation and phase separation were carried out as described for SHARK. The protein fraction was further concentrated using rotary evaporation and spray-dried. The swine hydrolysate (SWINE) was obtained from swine by-products (monospecies) collected from slaughterhouses in Portugal and processed according to Method 4 described in Annex IV, Chapter III of Regulation (EU) No 142/2011 of the EC. After processing, the resulting product was milled and hydrolyzed by ETSA (Empresa Transformadora de Subprodutos), under optimized temperature and pressure conditions, without the addition of chemicals (patent application PCT/IB2024/061806). The obtained protein fraction was further processed and spray-dried using the same method applied to the other hydrolysates. The enzyme concentrations and hydrolysis durations used in this study were selected based on extensive prior optimization work. These conditions were initially optimized at laboratory scale [17,26,27,28] and subsequently scaled up and validated at the industrial level at ETSA before being applied in the present study. A commercial fish protein hydrolysate (CPSP90) from Sopropêche, Wimille, France, was included as a high-value reference product.

### 2.2. Chemical Analysis

The proximate composition of hydrolysates was analyzed based on AOAC (Association of Official Analytical Chemists) recommendations—for dry matter (DM), ash, crude protein, crude fat, gross energy, and phosphorus contents—following procedures detailed in Marques et al. [29]. Briefly, DM was determined by oven drying at 105 °C until constant weight was achieved; ash content by incineration at 550 °C in a muffle furnace; nitrogen was determined by Dumas method and converted to crude protein using 4.9 [30] and 5.4 [31] conversion factors for SHARK and the remaining hydrolysates, respectively; crude fat by extraction using the Soxhlet method with petroleum ether; and gross energy was measured with a bomb calorimeter. Phosphorus content was determined colorimetrically using the Molybdenum Blue method after acid digestion. Chitin in INSECT was performed according to Guerreiro et al. [32], by spectrophotometric determination of glucosamine hydrochloride. Total phenolic content was also determined in INSECT, following procedures detailed in Monteiro et al. [33], by the Folin–Ciocalteu spectrophotometric method using 3,4,5-trihydroxybenzoic acid (gallic acid, Panreac Química S.L.U., Castellar del Vallès, Spain) as standard.

### 2.3. Amino Acid Profile and Peptide Molecular Weight Distribution

The AA profile of hydrolysates was determined by ultra-performance liquid chromatography, after a sample hydrolysis with 6 M hydrochloric acid, during 48 h, at 116 °C, as reported in Teodósio et al. [34]. The molecular weight distribution of peptides in the hydrolysates was obtained by high-performance size exclusion chromatography following the methodology described by Fernandez-Cunha et al. [35].

### 2.4. Protein Solubility

Protein solubility was determined according to Dent et al. [36]. Three separate dispersions were prepared for each protein hydrolysate by dissolving 1% protein hydrolysate in deionized water. The mixtures were stirred continuously for 30 min before adjusting the pH to 3, 5, or 7. Subsequently, 1 mL aliquots of each dispersion were transferred into microcentrifuge tubes in triplicate and centrifuged at 4400× *g* for 10 min. The resulting supernatants were then diluted 1:10 and analyzed for soluble protein concentration using the bicinchoninic acid (BCA) assay. The Pierce™ BCA Protein Assay Kit (Thermo Scientific, Waltham, MA, USA) was utilized according to the manufacturer’s protocol. A microplate assay was conducted by adding 25 μL of either the protein sample or standard into individual wells, followed by the addition of 200 μL of the working reagent. After thorough mixing, the plate was incubated at 37 °C for 30 min, then allowed to cool to room temperature (25 °C). Absorbance measurements were taken at 562 nm. Soluble protein concentration was determined using a bovine serum albumin standard curve ranging from 0 to 1 mg mL^−1^. The solubility of the protein was expressed as the ratio of the soluble protein concentration in the supernatant to the total crude protein of the hydrolysate used in the initial dispersion.

### 2.5. Antioxidant Activity

Before analysis, protein hydrolysates were resuspended in a 1:1 mixture of phosphate-buffered saline and methanol at a concentration of 25 mg mL^−1^. The 2,2′-azino-bis(3-ethylbenzothiazoline-6-sulfonic acid) (ABTS^•+^) and 2,2-diphenyl-1-picrylhydrazyl (DPPH^•^) radical scavenging activities of the hydrolysates were evaluated following the procedures described by Monteiro et al. [33], while the oxygen radical absorbance capacity (ORAC) assay was performed as detailed in Ribeiro et al. [37]. All assays were conducted using 96-well microplates, and the results were expressed as µmol Trolox eq g DM^−1^. Analyses were performed in triplicate (*n* = 3).

### 2.6. Impact on Bacterial Proliferation Dynamics

Bacterial strains with relevance for aquaculture and the media in which they were cultivated are summarized in Table 1.

All bacterial strains were obtained from the BCCM/LMG Bacteria Collection and grown aerobically on Tryptic soy broth (TSB; Oxoid, UK) with the exception of *Tenacibaculum maritimum* and *Vibrio harveyi*, which were grown in Marine Broth (Condalab, Madrid, Spain), at 25 °C, with agitation. A growth monitoring assay was conducted in 96-well flat-bottom polystyrene plates over 24 h. Protein hydrolysates were dissolved in each bacteria culture medium to a final concentration of 40 mg mL^−1^ and filtered through 0.22 µm nylon filters (VWR, Radnor, PA, USA) prior to the assay. The subsequent steps were performed as detailed in Monteiro et al. [38]. Briefly, 100 µL of each hydrolysate solution was added to 100 µL of each bacterial culture (OD 600 nm~0.1) in total volume of 200 µL, corresponding to a concentration of 20 mg mL^−1^ and a total assay biomass of 2 mg. A positive control (200 µL of diluted bacterial cultures—OD600~0.1) and a negative control (200 µL of culture medium) was also prepared. Growth was monitored by measuring optical density at 600 nm using a microplate reader (ELx808, Bio-Tek Instruments, Winooski, VT, USA). All growth data were normalized using a positive control, representing the normal growth of each bacterial strain incubated solely in culture media. The bacterial proliferation dynamics of each strain incubated with the protein hydrolysates were then compared to those observed when incubated with the CPSP90 hydrolysate.

### 2.7. Statistical Analysis

Antioxidant activity was subjected to one-way analysis of variance (ANOVA), with hydrolysate as main factor, after confirming normality (Kolmogorov–Smirnov test, *p* > 0.05) and homogeneity of variances (Levene’s test, *p* > 0.05). For protein solubility, a two-way ANOVA was performed with pH and hydrolysate as main factors. Repeated-measures ANOVA was used to assess the impact of the protein hydrolysates on bacterial growth dynamics over a period of 24 h. The probability level for rejection of the null hypotheses was set at 0.05. All statistics were performed using IBM SPSS Statistics 21.0 software (SPSS Inc., Chicago, IL, USA).

## 3. Results

### 3.1. Hydrolysates Proximate Composition

Hydrolysates’ chemical composition is represented in Table 2. The animal by-product hydrolysates had a high crude protein content, varying from 80.5% in FISH to 89.5% in SHARK. However, INSECT exhibited a lower protein content with 55.1%DM. Crude fat was present in low proportions (<9%) in all hydrolysates, with CPSP90 exhibiting the highest (8.4%DM) and FISH the lowest (0.1%DM) fat level. Overall, this resulted in similar energy levels among hydrolysates (19.4–20.8 kJ g^−1^ DM). The ash content ranged from 5.4 to 11.2% in SWINE and INSECT, respectively. Phosphorus levels were below 1% in all hydrolysates, ranging between 0.2%DM in SWINE and 0.7%DM in CPSP90. Chitin was also determined in INSECT, representing 0.04%DM.

### 3.2. Amino Acid Profile and Peptide Molecular Weight Distribution

The full amino acid profile of the hydrolysates is presented in Table 3.

Regarding the amino acids (AAs) composition, SHARK had the lowest total of essential amino acids (EAAs; 30.9 g 100 g^−1^ AAs), while CPSP90 and FISH exhibited the highest relative EAAs content (44.4 and 44.0 g 100 g^−1^ AAs, respectively). The lowest non-essential amino acids (NEAAs) relative content was observed in CPSP90 and FISH, resulting in the lowest NEAAs/EAAs ratios in these hydrolysates (1.3). In contrast, the highest total of NEAAs was observed in SHARK and SWINE (69.1 and 65.8 g 100 g^−1^, respectively). Collagenic AAs were most abundant in SHARK (46.7 g 100 g^−1^) and lowest in CPSP90 (23.6 g 100 g^−1^ AAs). Aromatic AAs were most abundant in INSECT (12.3 g 100 g^−1^ AAs) and lowest in SHARK (4.6 g 100 g^−1^ AAs). INSECT also had the highest content of branched-chain amino acids (BCAAs; 18.3 g 100 g^−1^), whereas SHARK showed the lowest (9.8 g 100 g^−1^ AA). Sulfur-containing amino acids were highest in CPSP90 (4.0 g 100 g^−1^ AA), followed closely by FISH (3.6 g 100 g^−1^ AA), and were lowest in SWINE (1.6 g 100 g^−1^ AA).

The molecular weight distribution of the tested hydrolysates is presented in Figure 1. SHARK was mainly composed of low- to medium-molecular-weight peptides (64%; 1 kDa < MW < 3 kDa), followed by medium-sized peptides (16%; 3 kDa < MW < 5 kDa). INSECT and FISH contained the highest proportion of small-molecular-weight peptides (35–36%; <1 kDa) and were mainly composed of low- to medium-molecular-weight peptides (51–57%; 1 kDa < MW< 3 kDa). In contrast, CPSP90 and SWINE exhibited a more uniform distribution of peptides across different molecular weight ranges.

### 3.3. Protein Solubility

In terms of protein solubility, an interaction between the hydrolysate and pH tested was observed (Figure 2). Across all pH values, INSECT was the only hydrolysate to consistently exhibit a higher protein solubility than CPSP90 (61–65% vs. 48–49%), regardless of pH. Conversely, SWINE showed a comparable solubility to CPSP90 regardless of pH (50–54%). The FISH solubility was equivalent to CPSP90 at pH 3 (42.9%), but lower at higher pH levels. In contrast, SHARK at pH 3 and pH 5 (26–28%) displayed the lowest protein solubility, with a slight improvement observed at pH 7 (36%).

### 3.4. Antioxidant Activity

The antioxidant capacity of the protein hydrolysates is presented in Figure 3. The ABTS^•+^ radical scavenging activity was highest in INSECT (469.9 µmol Trolox eq g DM^−1^), followed by FISH, and SWINE, all of which exhibited a higher activity compared to CPSP90. SHARK (175.9 µmol Trolox eq g DM^−1^) showed no significant difference from CPSP90. Similarly, the ORAC was significantly higher in INSECT and FISH (591.4–652.5 µmol Trolox eq g DM^−1^) compared to CPSP90, which showed no significant difference from SHARK and SWINE. No DPPH^•^ radical scavenging activity was detected in any of the samples.

### 3.5. Bacterial Growth Dynamics

In the growth monitoring assay, a significant inhibition of *Vibrio parahaemolyticus* was observed when incubated with SWINE, compared to CPSP90 (Figure 4). In contrast, FISH and SHARK significantly promoted the growth of this strain. Similarly, SWINE significantly inhibited *Vibrio anguillarum*, while INSECT significantly enhanced its growth. *Tenacibaculum maritimum* and *Vibrio harveyi* exhibited a similar behavior, their growth being inhibited by INSECT and enhanced by FISH and SHARK. For *Aeromonas salmonicida*, both SWINE and INSECT significantly inhibited its growth compared to CPSP90, whereas the other hydrolysates promoted its proliferation.

None of the tested hydrolysates inhibited *Pseudomonas aeruginosa*, with INSECT significantly enhancing its growth. Conversely, *Photobacterium damselae* subsp. *damselae* showed a significant growth increase when exposed to SHARK and FISH. No significant effects were observed for *Aeromonas hydrophila* subsp. *hydrophila* when incubated with the experimental protein hydrolysates.

## 4. Discussion

The analysis of protein hydrolysates revealed significant differences in their chemical composition, functional properties, and bioactivity, emphasizing the unique characteristics and potential applications of each hydrolysate produced. A key determinant of these differences was the hydrolysis method employed, with SHARK, FISH, and INSECT produced via enzymatic hydrolysis using alcalase, while SWINE was processed through a chemical-free method involving temperature and pressure. These two approaches revealed distinct variations in the peptide profile, solubility, and bioactivity, which in turn influenced their suitability for different applications.

Most hydrolysates exhibited a high crude protein content, with SHARK and SWINE showing the highest values. This is consistent with their origins from high-protein raw materials, such as collagen-rich tissues, like shark skin [17] and swine meat and bone [39]. Typically, the protein content of protein hydrolysates from different sources ranges from 60% to 90% of the total composition, which makes them valuable protein sources for food and feed applications [26,40,41,42]. The low crude fat content (<5%) observed in most hydrolysates, except for CPSP90, can be attributed to the phase separation process used during their production. This is relevant as the retention of a high amount of fat in the final products may limit the use of ingredients in food/feed applications, due to high lipid contents leading to an undesirable taste and darkened color due to changes in the lipids [43]. Lipid residues in protein hydrolysates must be lower than 0.5% in order to prevent the negative alteration of the lipids during storage [44]. In the case of insect-derived hydrolysates, chitin has been identified as a potential allergen that could negatively affect their nutritional value and safety, as well as the health and product quality of animals consuming these feeds [45]. In insect larvae meals it usually ranges from 5 to 7% DM, varying with insect species and its processing [46,47]. In the present work, hydrolysis effectively minimized chitin levels to less than 0.5%DM, which could enhance the hydrolysate’s suitability for aquafeed and other animal nutrition applications by improving digestibility and reducing allergenic risks

The AA composition of the hydrolysates revealed distinct patterns that influence their nutritional value and functionality. FISH and CPSP90 exhibited the highest relative EAAs content and a similar overall AA profile to high-quality fishmeal [48]. This supports the use of protein hydrolysates based on marine by-products for improved growth in aquatic and terrestrial species [49,50], guaranteeing the strategic use of fishmeal as a feed ingredient and safeguarding the sustainability of feed formulations, particularly those directed for aquaculture [51,52]. On the other hand, SHARK, with its high levels of glycine, proline, and hydroxyproline, reflects a collagen-rich composition, which is consistent with other studies on fish skin hydrolysates [17]. While these AAs are deemed non-essential, their adequate dietary supplementation has been reported to be essential for maximum collagen synthesis and improved animal growth and health [53]. The AAs profile of hydrolysates also directly impacts their antioxidant properties, as specific amino acids contribute to free radical scavenging and oxidative defense mechanisms [54]. INSECT, which demonstrated the highest ABTS^•+^ and ORAC activities, is characterized by a rich relative content in aromatic AAs, such as tyrosine and tryptophan. These AAs are well-documented for their ability to donate protons and neutralize reactive oxygen species (ROS), stabilizing free radicals in the process [54]. However, other hydrolysates rich in aromatic AAs, like CPSP90, did not exhibit as high an antioxidant activity, suggesting that additional compounds in INSECT may also have contributed to its strong antioxidant potential. Since chitin levels are residual (<0.5%), it is more likely that phenolic compounds associated with *Hermetia illucens* feeding regimes may have contributed to the antioxidant activity observed. As this species is often reared in vegetable and plant by-products, it is likely to have accumulated phenolic compounds retrieved from these substrates as reported by Oh et al. [55]. Indeed, the phenolic compound content in INSECT was, overall, more than double of what has been reported in other insect species [56,57]. On the other hand, the small peptide fraction in INSECT (<1 kDa, 36%) and FISH further support their higher antioxidant potential by providing easily accessible bioactive residues for ROS interactions [58]. On the other hand, while the uniform peptide distribution in CPSP90 and SWINE may enhance their versatility in applications, it may hinder interactions with ROS, contributing to the lower antioxidant capacity observed. In contrast, the low antioxidant activity of SHARK may be attributed to its high content of collagenic Aas, such as glycine, proline, and hydroxyproline. These AAs are less soluble in aqueous environments and tend to aggregate or interact with lipids [59], making them less effective as antioxidants. Furthermore, they lack aromatic or sulfhydryl groups, which are key radical scavengers, further limiting their ability to counteract ROS [60]. Nonetheless, all tested protein hydrolysates exhibited a higher antioxidant capacity than values reported for plant-based protein sources [61]. Moreover, those with highest activities (INSECT and FISH) displayed antioxidant levels comparable to those of antioxidant-rich fruits and vegetables, such as berries [62,63]. This could be particularly relevant to improve the protection against oxidative stress and improve the stability of these proteins and the food systems in which they are incorporated. While the majority of the hydrolysates exhibited a low lipid content (<3%) and high protein content (>55%), we acknowledge that the oxidative stability was not evaluated. Future studies should assess both lipid peroxidation and protein oxidation—such as carbonyl formation—over time to better inform the shelf-life, storage potential, and overall product quality

Among the functional properties of proteins, solubility is of the most important, due to its significant influence on the interfacial properties of proteins and peptides [64]. In the present work, INSECT exhibited the highest solubility across all pH levels tested. Generally, proteins exhibit the highest solubility in alkaline pH ranges, where the pH is well above their isoelectric point [65]. Similarly to plant proteins, we hypothesize most hydrolysates in this study may exhibit isoelectric points around pH 4–6 where aggregation is common due to weak electrostatic repulsion. This hypothesis is supported by the amino acid composition of the hydrolysates, particularly their high content of aspartic and glutamic acids, which have side chain pKa values near 3.4 and 4.1, respectively [66]. While charged side chains typically enhance solubility, the present study found no direct correlation between the hydrophilic amino acid content and solubility, as INSECT exhibited the highest solubility amongst the hydrolysates but not the highest content of hydrophilic amino acids. In contrast, the low solubility of SHARK likely reflects the hydrophobic nature of collagenic amino acids, which reduces their interaction with water and may lead to aggregation [67,68]. Other components such as sugars, minerals, and lipids may also have influenced the solubility, particularly in these hydrolysates with a lower protein content, such as INSECT. Typically, the solubility of insect-meal-based proteins is low (<30%), which might be partly related to the intensive processing they undergo [67]. Although extraction, fractionation, and enzymatic hydrolysis can improve solubility, as reported values for *H. illucens* hydrolysates generally remain below 60% across pH 3–9. However, in the present work, the INSECT solubility ranged from 61 to 65%, making it comparable to high-quality protein sources such as fishmeal [69]. This is important when considering its incorporation into aquafeeds. But while a high solubility is desirable to enhance protein availability, excessively high values may increase leaching, potentially compromising both the protein quality and retention in aquafeeds [70]. In fact, Tonheim et al. [69] reported that inclusion levels exceeding 15% of the hydrolyzed protein in aquafeeds led to significantly higher leaching compared to non-hydrolyzed protein, ultimately hindering diet digestibility. In another study by Tonheim et al. [71], it was demonstrated that increasing the proportion of water-soluble nitrogen can improve but at times can also hinder or not at all affect ingredient digestibility. In parallel, the interaction with other ingredients was shown to affect overall diet digestibility, possibly due to the protease inhibition from other ingredient constituents. Therefore, the impact of hydrolysates’ protein solubility in diet digestibility will be likely conditioned by the basal formulation they are included in.

Given their intrinsic effect on the amino acid profile and chemical composition, the conditions of hydrolysis have likely played a key role in protein solubility. While enzymatic hydrolysis is known to significantly enhance solubility [64,72], in this study the highest solubility was observed in the INSECT hydrolysate followed by SWINE which were obtained by using a chemical-free process based on temperature and pressure. As demonstrated in this work, there is a distinct difference in the peptide sizes generated via the enzymatic hydrolysis with alcalase (SHARK, FISH, and INSECT) compared to the chemical-free SWINE. Beyond differences in the peptide molecular weight, variations in solubility are likely attributed to the formation of peptides with different isoelectric points [72], which in the majority of tested hydrolysates appear to be charged within the pH range of 3–7. In contrast, the apparent increase in the protein solubility of SHARK at pH 7 suggests its isoelectric point is lower, while the opposite is true for FISH. Although solubility measurements in this study were limited to pH 3, 5, and 7, these values suggest that the proteins likely remain charged within this range. To confirm the precise isoelectric points, future studies should assess solubility over a more refined pH spectrum. Following hydrolysis, all protein hydrolysates were dried into a powder to enable long-term storage and facilitate their incorporation into different food systems. Therefore, the drying process is another factor affecting protein solubility, and it depends on the intensity and duration of the heat treatment. Poor solubility can also be derived from the formation of the crust film upon the unfolding of one to several protein molecules at the interface between the particle and air during drying, particularly spray-drying [73]. In fact, the solubility of protein hydrolysates produced at a pilot or production scale is usually lower than those of laboratory-scale hydrolysates due to the harsh drying conditions (e.g., spray-drying), which alter the native structure of protein molecules [73]. This supports the low to moderate solubility (26–65%) observed among all of the experimental hydrolysates that were produced at pilot-scale conditions.

The effect of protein hydrolysates on bacterial proliferation was broad, varying across bacterial strains. In particular, SWINE significantly inhibited the growth of multiple strains compared to CPSP90, suggesting that its inclusion in aquafeeds may help reduce or delay the proliferation of pathogenic bacteria such as *A. salmonicida* and specific *Vibrio* strains, including *V. parahaemolyticus* and *V. anguilarum.* This finding aligns with reports that porcine-derived peptides possess specific antimicrobial properties, which are attributed to their ability to disrupt bacterial membranes [74]. These authors also found the highest activity in fractions containing medium (>5–10 kDa) and large peptides (>10 kDa), aligning with the peptide profile observed herein in SWINE. In addition, the high hydrophobicity of SWINE amino acids could have contributed to its antimicrobial activity, as a higher hydrophobicity aids in binding lipopolysaccharides on the bacteria’s outer cell membrane [75]. Although the inhibitory potential of this hydrolysate is not high enough for veterinary use, its prophylactic application in animals, particularly marine fish vulnerable to stressful conditions (e.g., transportation and handling) or early-stage infections, could enhance resilience by delaying bacteria proliferation and boosting antimicrobial efficacy. In addition to inhibiting bacterial growth, the protein hydrolysates also promoted the growth of opportunistic bacterial, such as *A. salmonicida, T. maritimum, V. harveyi*, and *V. parahaemolyticus*, particularly with FISH and SHARK. INSECT also promoted the growth of *V. anguliarum* and *P. aeruginosa*, highlighting potential risks associated with certain hydrolysates. The small peptide size may have facilitated membrane penetration, enhancing access to amino acids and promoting bacterial growth [76]. While this may pose a concern in certain contexts, it also suggests the potential for hydrolysates as components in bacterial culture media. Hydrolysates rich in small peptides and EAAs, such as SHARK, FISH, and INSECT, may be particularly suitable for this application [4,77]. Their composition can provide a readily available source of nitrogen and carbon for microbial fermentation, supporting high bacterial growth as shown by Kuo et al. [78], who reported the high growth of aquatic pathogens, including *A. hydrolphyla,* in the presence of a fish protein hydrolysate derived from tilapia by-products. However, these hydrolysates may also prove useful in the cultivation of probiotics, which are often known to exclude pathogens by competitive exclusion [78,79]. Thus, the selective growth-promoting properties of certain hydrolysates could be leveraged to cultivate beneficial bacterial strains or modulate microbial communities in aquaculture systems [80].

## 5. Conclusions

This study highlights the diverse chemical composition, functional properties, and bioactivities of protein hydrolysates derived from processed animal proteins, reinforcing their potential in aquaculture and functional feed applications. The protein content, amino acid profile, bioactivities, and solubility varied significantly with the source material and processing conditions. FISH and CPSP90 exhibited the highest essential amino acid content, while SHARK and SWINE hydrolysates were rich in collagen-derived amino acids. FISH and INSECT exhibited the greatest antioxidant capacities, likely due to their high proportions of low-molecular-weight peptides, aromatic amino acids, and, in the case of INSECT, phenolic compounds. INSECT also demonstrated the highest solubility across all tested pH levels and, along with SWINE, exhibited mild and selective antimicrobial activity, suggesting a potential in disease mitigation. In contrast, FISH and SHARK tended to support the growth of opportunistic bacteria, indicating possible applications as nitrogen sources in microbial culture media. Future research should focus on identifying specific bioactive peptides, elucidating their mechanisms against ROS and pathogens, and assessing their long-term impacts on animal health and performance.

## Figures and Tables

**Figure 1 antioxidants-14-00782-f001:**
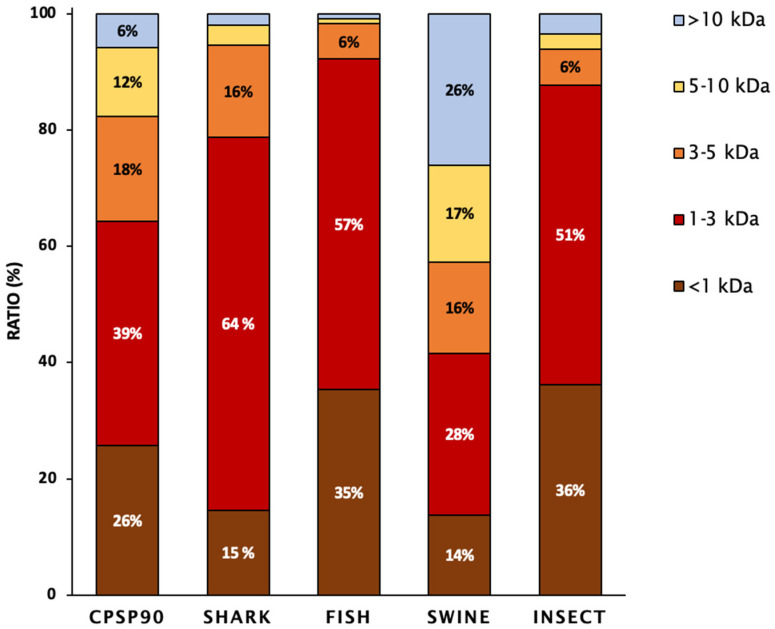
The peptide molecular weight distribution of the protein hydrolysates, expressed as the percentage of total peptides. Values represent the proportion of each molecular weight fraction relative to the total peptide mass recovered from the hydrolysate.

**Figure 2 antioxidants-14-00782-f002:**
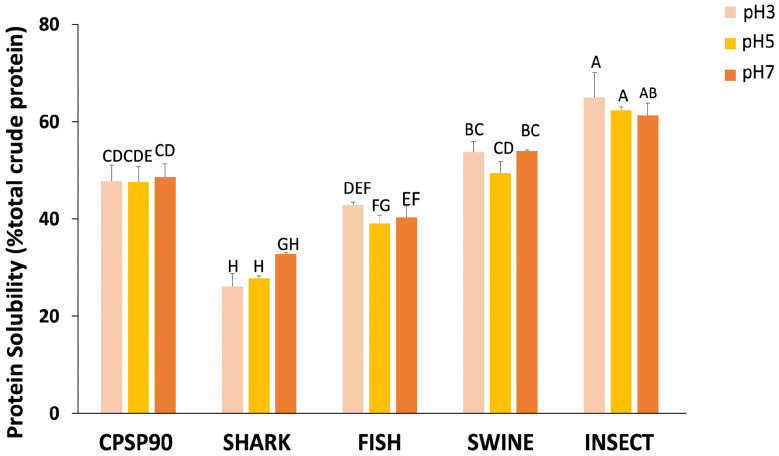
The protein solubility (% total crude protein) of the experimental protein hydrolysates at pH 3, 5, and 7. Different capital letters indicate statistical differences in the interaction between pH and the hydrolysate (*p* = 0.010).

**Figure 3 antioxidants-14-00782-f003:**
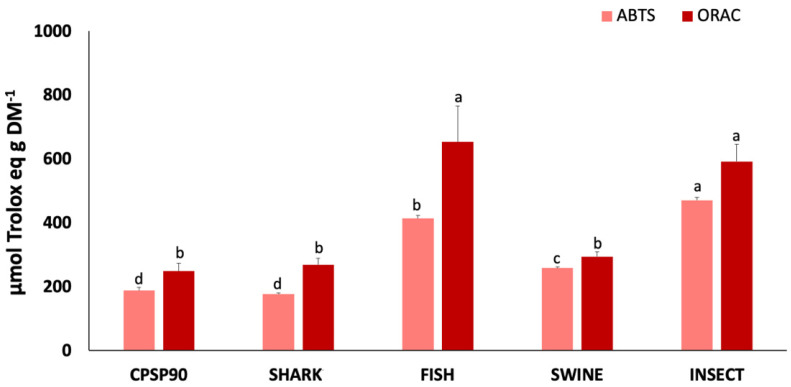
2,2′-azino-bis(3-ethylbenzothiazoline-6-sulfonic acid) (ABTS^•+^) radical scavenging activity and oxygen radical absorbance capacity (ORAC) of protein hydrolysates (*n* = 3). Different lowercase letters indicate statistical differences between hydrolysates.

**Figure 4 antioxidants-14-00782-f004:**
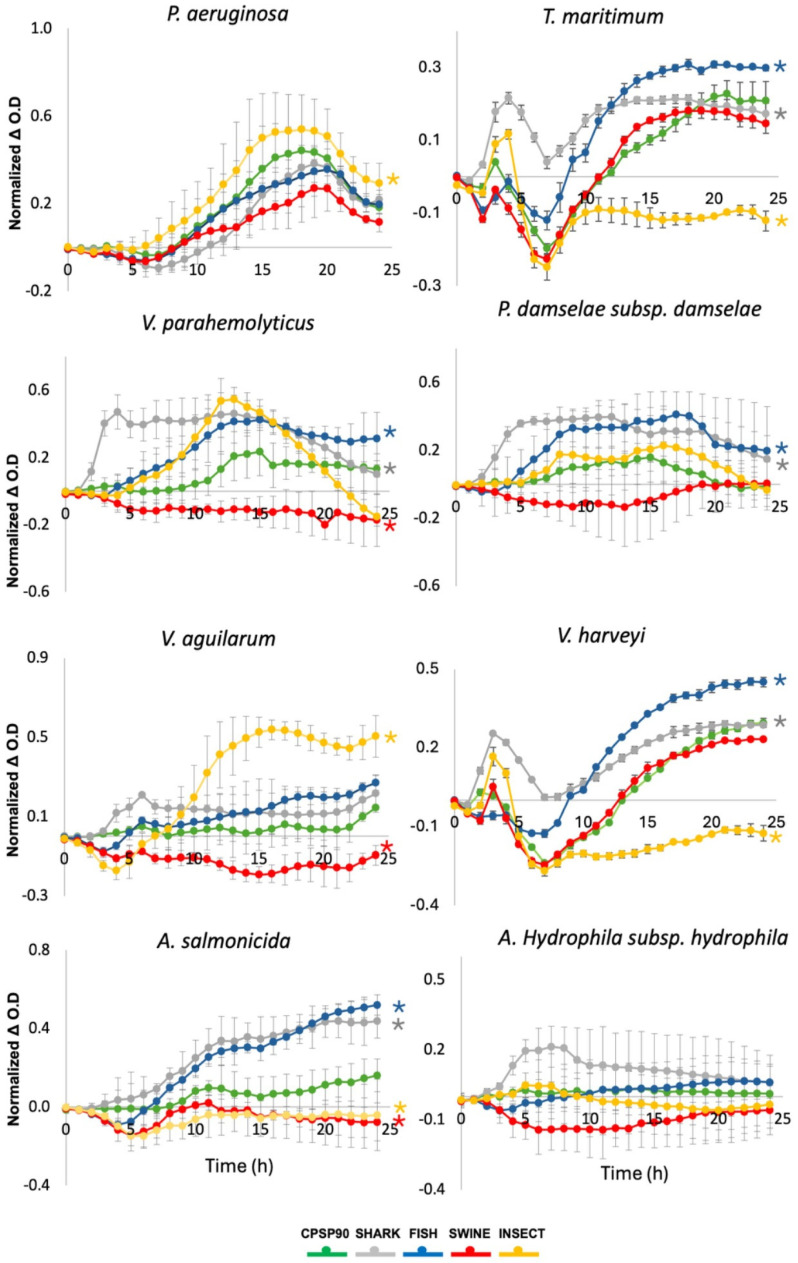
Bacterial growth dynamics of pathogenic strains incubated with the experimental protein hydrolysates, during 24 h. Growth curves were calculated by subtracting the OD600 nm of bacteria incubated with each hydrolysate from the OD600 nm of bacteria grown in the medium alone. Asterisks indicate statistically significant differences (*p* < 0.05) compared to CPSP90.

**Table 1 antioxidants-14-00782-t001:** Bacterial strains and culture media used for growth monitoring assay.

Bacterial Strains	Culture Media
*Aeromonas hydrophila* subsp. *hydrophila* (LMG 2844)	TSB
*Aeromonas salmonicida* (LMG 3780)	TSB
*Pseudomonas aeruginosa* (LMG 12228)	TSB
*Photobacterium damselae* subsp. *damselae* (LMG 19445)	TSB
*Vibrio anguillarum* (LMG 4437)	TSB
*Vibrio parahaemolyticus* (LMG 2850)	TSB
*Vibrio harveyi* (LMG 18298)	MB
*Tenacibaculum maritimum* (LMG 13040)	MB

Bacterial strains were obtained from bacterial collections (BCMZ/LMG, Belgian Coordinated Collections of Microorganisms, Laboratory of Microbiology, Department of Biochemistry and Microbiology, Faculty of Sciences of Ghent University, Ghent, Belgium).

**Table 2 antioxidants-14-00782-t002:** Protein hydrolysates proximate composition (%DM).

	CPSP90	SHARK	FISH	SWINE	INSECT
Dry Matter (DM)	96.2	93.6	94.0	95.6	98.0
Crude Protein	83.7	89.5	80.5	87.6	55.1
Crude Fat	8.4	0.34	0.1	0.2	2.0
Energy (kJ g^−1^ DM)	19.6	19.4	20.8	20.5	19.7
Ash	7.9	5.8	6.4	5.4	11.2
Phosphorus	0.7	0.2	0.6	0.2	0.5
Chitin	-	-	-	-	0.2
Total Phenolics (gallic acid eq.)	-	-	-	-	2.7

**Table 3 antioxidants-14-00782-t003:** The amino acid (AA) quantitative profile of the protein hydrolysates (g 100 g^−1^ AA).

	CPSP90	SHARK	FISH	SWINE	INSECT
**Essential amino acids (EAA)**					
Arginine	6.8	7.5	5.0	6.6	4.0
Histidine	2.2	1.3	2.1	2.1	3.4
Lysine	7.9	4.7	8.3	5.5	5.7
Threonine	4.2	2.9	4.3	2.8	3.5
Isoleucine	3.8	2.8	4.1	2.5	4.3
Leucine	7.2	3.9	7.6	5.6	6.8
Valine	4.6	3.0	5.1	3.9	7.2
Methionine	2.9	2.2	2.9	1.4	1.3
Phenylanine	4.0	2.3	3.9	3.2	3.3
Tryptophan	0.9	0.2	0.8	0.5	1.4
**Σ EAA**	44.4	30.9	44.0	34.2	41.0
**Non-essential amino acids (EAA)**					
Cysteine	1.0	0.3	0.7	0.2	0.8
Tyrosine	2.9	0.7	1.6	1.9	4.2
Aspartic acid + asparagine	9.3	6.0	9.5	6.5	8.8
Glutamic acid + glutamine	13.7	11.0	15.6	13.3	14.8
Alanine	6.9	9.0	7.8	8.4	11.1
Glycine	9.8	20.7	9.3	16.1	6.7
Proline	5.2	10.0	5.4	9.5	7.5
Serine	5.0	3.8	3.7	3.2	3.7
Hydroxyproline	1.7	6.9	1.7	6.2	0.4
Ornithine	-	0.4	0.8	0.4	1.1
**Σ NEAA**	55.6	69.1	56.0	65.8	59.0
**NEAA/EAA**	1.3	2.2	1.3	1.9	1.4
**Amino acid Classes**					
Positive	16.9	13.9	16.2	14.6	14.3
Hydrophilic	53.0	38.8	51.6	42.6	50.1
Collagenic	23.6	46.7	24.1	40.2	25.7
Aromatic	10.0	4.6	8.4	7.7	12.3
Branched-chain	15.6	9.8	16.8	12.1	18.3
Sulfur	4.0	2.5	3.6	1.6	2.0

Positive AAs are the sum of lysine, arginine, histidine, and ornithine. Hydrophilic AAs are the sum of arginine, histidine, lysine, threonine, cystine, tyrosine, aspartic acid + asparagine; glutamic acid + glutamine, serine, and ornithine. Collagenic AAs are the sum of proline, alanine, glycine, and hydroxyproline. Aromatic AAs are the sum of histidine, tryptophan, tyrosine, and phenylalanine. Branched-chain AAs are the sum of leucine, isoleucine, and valine. Sulfur AAs are the sum of methionine and cystine.

## Data Availability

Data will be made available upon request.

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
