# Peer review of "Circular Animal Protein Hydrolysates: A Comparative Approach of Functional Properties"

_antioxidants, 2025, doi:10.3390/antiox14070782_

Round 1

Reviewer 1 Report

I provided my comments below.

The manuscript by Monteiro et al. entitled “Circular Animal Protein Hydrolysates: a comparative approach of functional properties” is a research article in which Authors investigated important functional properties such as antioxidant activity and modulation of microbial growth of newly prepared protein hydrolysates. The test protein hydrolysates were obtained from processed animal by-products including insects, shark, fish, and porcine, in upcycling processes utilising hydrolysis. This research represents modern attitude within sustainable development, which is important challenge in modern world. 
The article is interesting and well written.
Please find below my minor comments:
1. The source of material used for preparation of “FISH” and “SWINE” should be provided similarly to that for preparation of “SHARK” and “INSECTS” (eg. which company provided the material?”). 
2. Page 4, line 144 - what o you mean by “diets” in this sentence: “The AAs profile of hydrolysates and diets was determined” ?

Author Response

The authors express their sincere gratitude to both reviewers for their valuable time and thoughtful engagement in reviewing the manuscript. Their careful reading and constructive feedback have been instrumental in enhancing the clarity and depth of the research. We genuinely appreciate their insightful comments and suggestions, which have significantly improved the quality of the manuscript. We have made diligent efforts to address the raised concerns and incorporate the recommended revisions. The reviewers’ feedback has helped refine the content, and ensure the manuscript meets the standards for publication.

Reviewer #1

 The manuscript by Monteiro et al. entitled “Circular Animal Protein Hydrolysates: a comparative approach of functional properties” is a research article in which Authors investigated important functional properties such as antioxidant activity and modulation of microbial growth of newly prepared protein hydrolysates. The test protein hydrolysates were obtained from processed animal by-products including insects, shark, fish, and porcine, in upcycling processes utilising hydrolysis. This research represents modern attitude within sustainable development, which is important challenge in modern world. 
The article is interesting and well written.
Please find below my minor comments:

1.The source of material used for preparation of “FISH” and “SWINE” should be provided similarly to that for preparation of “SHARK” and “INSECTS” (eg. which company provided the material?”). 

Thank you for the comment. The collection of the shark, fish, insect, and swine by-products were performed by ETSA-SGPS, S.A. from Portuguese suppliers. Specifically:

  • The FISH hydrolysate originates from multispecies by-products sourced from various retailers and distributors in the fish processing and canning industries.
  • The SWINE hydrolysate is derived from monospecies by-products obtained from several slaughterhouses throughout the country.

We have revised the manuscript (lines 109-110 and lines 122-124) to clarify this multi-source and multi-species origin, ensuring transparency while reflecting the practical realities of industrial-scale by-product collection. The following text has been added:

“The fish hydrolysate (FISH) was produced from fish by-products (multispecies) obtained from retailers and distributors in the fish processing and canning industries in Portugal.”

“The swine hydrolysate (SWINE) was obtained from swine by-products (monospecies) collected from slaughterhouses in Portugal, and processed according to Method 4 described in Annex IV, Chapter III of Regulation (EU) No 142/2011 of the EC.”

2. Page 4, line 144 - what o you mean by “diets” in this sentence: “The AAs profile of hydrolysates and diets was determined” ?

Thank you for your comment. This was typo, so the sentence has been re-written as “The AAs profile of hydrolysates was determined”.

Reviewer 2 Report

This manuscript aimed to report the results of a study which evaluated functional and bioactive properties of five novel protein hydrolysates derived from various animal by-products (shark skin, fish by-products, swine by-products, and insect larvae meal) by comparing them to the commercial hydrolysate CPSP90.  While the study provides useful insight into the functional properties of animal protein hydrolysates, it would benefit from clearer framing that these properties are preliminary indicators rather than direct evidence of efficacy in live animals. In my opinion, I would feel more confident is more work was included to validate the  assumptions made by the authors for expected benefits of using these protocol for protein hydrolysates for animal feed.

These are some speficic issues I have with this study:

  1. The authors reference a patent but do not provide sufficient details regarding the SWINE hydrolysate processing conditions. This lack of detail limits reproducibility and validation by other researchers.
  2. The rationale behind selecting specific enzyme concentrations and the different durations of hydrolysis should be explicitly justified. Without such rationale, interpreting observed differences in functional properties becomes less meaningful.
  3. The microbial inhibition assays used a relatively high concentration (40 mg/mL) without clear justification or practical relevance. Such a concentration may exceed realistic dietary inclusion levels, limiting practical implications of the findings.
  4. It is unclear from the description whether appropriate controls (positive and negative) were used in microbial growth assays. Clearly defined control conditions are essential for accurately interpreting the impact of the hydrolysates on bacterial growth.
  5. The manuscript does not report data on lipid oxidation. Evaluating lipid oxidation is important for assessing shelf-life stability, a critical parameter for the practical application of these protein hydrolysates.

Author Response

The authors express their sincere gratitude to both reviewers for their valuable time and thoughtful engagement in reviewing the manuscript. Their careful reading and constructive feedback have been instrumental in enhancing the clarity and depth of the research. We genuinely appreciate their insightful comments and suggestions, which have significantly improved the quality of the manuscript. We have made diligent efforts to address the raised concerns and incorporate the recommended revisions. The reviewers’ feedback has helped refine the content, and ensure the manuscript meets the standards for publication.

Reviewer #2

This manuscript aimed to report the results of a study which evaluated functional and bioactive properties of five novel protein hydrolysates derived from various animal by-products (shark skin, fish by-products, swine by-products, and insect larvae meal) by comparing them to the commercial hydrolysate CPSP90. While the study provides useful insight into the functional properties of animal protein hydrolysates, it would benefit from clearer framing that these properties are preliminary indicators rather than direct evidence of efficacy in live animals. In my opinion, I would feel more confident is more work was included to validate the assumptions made by the authors for expected benefits of using these protocol for protein hydrolysates for animal feed.

1.The authors reference a patent but do not provide sufficient details regarding the SWINE hydrolysate processing conditions. This lack of detail limits reproducibility and validation by other researchers.

The swine hydrolysate (SWINE_H) was obtained from swine by products (collected from slaughterhouses in Portugal) processed according to Method 4 described in Annex IV, Chapter III of Regulation (EU) No 142/2011 of the EC as mentioned in the M&M section. The resulting product was milled with an industrial chopper to achieve a homogeneous mixture and then subjected to optimised pressure and temperature conditions established by ETSA-SGPS, S. A., without the addition of chemicals (Patent Application PCT/IB2024/061806). The detailed processing conditions are under a patent application, so no further details can be provided. Moreover, this product will soon be produced at industrial scale by ETSA, being available in the market. So, at that point they can be further tested by other researchers.

2.The rationale behind selecting specific enzyme concentrations and the different durations of hydrolysis should be explicitly justified. Without such rationale, interpreting observed differences in functional properties becomes less meaningful.

Thank you for the comment. The enzyme concentrations and hydrolysis durations used in this study were selected based on extensive prior optimization work. These conditions were initially optimized at laboratory scale (Borges et al. 2022a, 2022b, 2023; Coscueta et al., 2024) and subsequently scaled up and validated at the industrial level at ETSA

We have updated the manuscript to include this information along with relevant references, in lines 127-131 “The enzyme concentrations and hydrolysis durations used in this study were selected based on extensive prior optimization work. These conditions were initially optimized at laboratory scale [17, 26-28], and subsequently scaled up and validated at the industrial level at ETSA before being applied in the present study.”

Borges, Sandra, et al. "Production of insect protein hydrolysates: a multifactorial study." 1st International Congress on Food, Nutrition and Public Health. (2022a).

Borges, Sandra, et al. "Valorization of porcine by-products: a combined process for protein hydrolysates and hydroxyapatite production." Bioresources and Bioprocessing 9.1 (2022b): 30.

Borges, Sandra, et al. "Fish by-products: A source of enzymes to generate circular bioactive hydrolysates." Molecules 28.3 (2023): 1155.

Coscueta, Ezequiel R., et al. "Turning discarded blue shark (Prionace glauca) skin into a valuable nutraceutical resource: An enzymatic collagen hydrolysate." Food Bioscience 60 (2024): 104472.

3.The microbial inhibition assays used a relatively high concentration (40 mg/mL) without clear justification or practical relevance. Such a concentration may exceed realistic dietary inclusion levels, limiting practical implications of the findings.

We appreciate the reviewer’s concern regarding the concentration used in our microbial inhibition assays. Our assay started with hysolysates diluted to 40 mg/mL. Then, 100 ul of each hydrolysates was added to 100 ul of each bacteria in total volume of 200 µL, corresponding to a concentration of 20 mg/mL and a total assay biomass of 2 mg.

Hydrolysates are typically included at 3–5% (w/w) in aquafeeds, as reported in several studies (Khosravi et al. 2015; Fan et al. 2022; Resende et al. 2022). This inclusion rate translates to approximately 30–50 mg of hydrolysate per gram of feed. Given an average voluntary feed intake (VFI) of 1.5% body weight per day (0.015 g feed/g average body weight (ABW)/day) for European seabass (Monteiro et al. 2024; Fernandes et al. 2022) or gilthead seabream (Tampou et al. 2024), the estimated daily hydrolysate intake at 3% inclusion is approximately:

0.015 g feed/g ABW/ day×30 mg hydrolysate/g feed=0.45 mg hydrolysate/g ABW/day

Comparing these values, the 2 mg hydrolysate biomass used in the assay corresponds roughly to the amount a fish of 10-50 g would ingest in less than one day. Although the assay concentration appears higher on a per-volume basis, the dose is actually comparable to realistic daily intake levels when considering feed consumption. Furthermore, continuous feeding over several weeks ensures accumulation of effective doses, supporting the practical relevance of our in vitro antimicrobial findings during longer feeding trials.

In fact, in vivo trials using these hydrolysates have already been conducted and are currently being prepared for publication. We agree that in vivo validation is essential, but emphasize that this manuscript represents an initial screening of the potential of various hydrolysates. This approach was valuable for identifying the most promising candidates and optimizing experimental conditions—such as selecting the most effective bacterial challenge—for subsequent in vivo studies.

Fan, Ze, et al. "Assessment of fish protein hydrolysates in juvenile largemouth bass (Micropterus salmoides) diets: effect on growth, intestinal antioxidant status, immunity, and microflora." Frontiers in Nutrition 9 (2022): 816341.

Khosravi, Sanaz, et al. "Effects of protein hydrolysates supplementation in low fish meal diets on growth performance, innate immunity and disease resistance of red sea bream Pagrus major." Fish & shellfish immunology 45.2 (2015): 858-868.

Fernandes, Helena, et al. "Pre-treatment of Ulva rigida improves its nutritional value for European seabass (Dicentrarchus labrax) juveniles." Algal Research 66 (2022): 102803.

Monteiro, M., et al. "Towards sustainable aquaculture: Assessing polychaete meal (Alitta virens) as an effective fishmeal alternative in European seabass (Dicentrarchus labrax) diets." Aquaculture 579 (2024): 740257.

Resende, Daniela, et al. "Innovative swine blood hydrolysates as promising ingredients for European seabass diets: impact on growth performance and resistance to Tenacibaculum maritimum infection." Aquaculture 561 (2022): 738657.

Tampou, Anna, et al. "Growth performance of gilthead sea bream (Sparus aurata) fed a mixture of single cell ingredients for organic diets." Aquaculture Reports 36 (2024): 102105.

4.It is unclear from the description whether appropriate controls (positive and negative) were used in microbial growth assays. Clearly defined control conditions are essential for accurately interpreting the impact of the hydrolysates on bacterial growth.

We appreciate the reviewer’s attention to the importance of control conditions in microbial growth assays. In our experiments, we included two essential controls:

  • A positive control consisting of bacteria cultured in the standard growth medium without hydrolysate, to confirm normal bacterial growth.
  • A negative control containing only the culture medium without bacteria, to ensure sterility and rule out contamination.

These controls allowed us to accurately assess the impact of the hydrolysates on bacterial growth. We have clarified these details in the Methods section in lines 198-202.

“Briefly, 100 µl of each hydrolysate solution was added to 100 µl of each bacterial culture (OD 600nm ~ 0.1) in total volume of 200 µL, corresponding to a concentration of 20 mg mL-1 and a total assay biomass of 2 mg. A positive control (200 µL of diluted bacterial cultures – OD600 ~ 0.1), a negative control (200 µL of culture medium), was also prepared.

5.The manuscript does not report data on lipid oxidation. Evaluating lipid oxidation is important for assessing shelf-life stability, a critical parameter for the practical application of these protein hydrolysates.

We appreciate the reviewer’s suggestion regarding lipid oxidation analysis. As noted in the manuscript, the tested protein hydrolysates have a very high protein content (55.1-89.5%) , but a low lipid content (<3%), which is a significant advantage in terms of oxidative stability and shelf-life. The minimal lipid presence reduces susceptibility to lipid oxidation, thereby enhancing product stability during storage.

Given this low lipid level, lipid oxidation was considered of limited relevance for this particular study. However, we acknowledge its importance for hydrolysates or products with higher lipid content and will consider incorporating such analyses in future work.

Round 2

Reviewer 2 Report

I acknowledge the effort of the authors to address all my comments and suggestions. I am only asking one minor addition in the limitations of this study.

I appreciate the authors' response about lipid oxidation. While the hydrolysates have low lipid content, I believe lipid oxidation remains a crucial issue. Even lean fish, with similar low fat, are prone to oxidation due to unsaturated fatty acids and pro-oxidants. Stating lipid oxidation is "of limited relevance" without data creates a significant gap for practical application.  For this reason,  I recommend that the authors should acknowledge this as a study limitation in the discussion, clarifying that despite low lipid content, its impact on long-term stability wasn't directly assessed.

Author Response

I appreciate the authors' response about lipid oxidation. While the hydrolysates have low lipid content, I believe lipid oxidation remains a crucial issue. Even lean fish, with similar low fat, are prone to oxidation due to unsaturated fatty acids and pro-oxidants. Stating lipid oxidation is "of limited relevance" without data creates a significant gap for practical application. For this reason, I recommend that the authors should acknowledge this as a study limitation in the discussion, clarifying that despite low lipid content, its impact on long-term stability wasn't directly assessed.

We appreciate the reviewer’s comment and fully acknowledge the importance of lipid oxidation, even in materials with low fat content. Although the hydrolysates analyzed in this study had lipid levels below 3%, oxidation processes—both lipid peroxidation and protein oxidation—can still occur, particularly due to the presence of unsaturated fatty acids and pro-oxidant compounds. This is especially relevant considering that these hydrolysates also have high protein content (>55%), which may be prone to oxidative modifications such as protein carbonylation. Evaluating oxidative stability over time is therefore essential to better understand their shelf-life, safety, and functional performance in applied contexts.

In response to this, we have revised the Discussion section (lines 370–374) to acknowledge this limitation and to highlight the need for further studies. It now reads:

“While the majority of the hydrolysates exhibited low lipid content (<3%) and high protein content (>55%), we acknowledge that oxidative stability was not evaluated. Future studies should assess both lipid peroxidation and protein oxidation—such as carbonyl formation—over time to better inform shelf-life, storage potential, and overall product quality.”